# Patients with Subarachnoid Hemorrhage Exhibit Disturbed Expression Patterns of the Circadian Rhythm Gene *Period-2*

**DOI:** 10.3390/life11020124

**Published:** 2021-02-05

**Authors:** Sibylle Frase, Sandra Kaiser, Matti Steimer, Lisa Selzner, Niels Alexander Foit, Wolf-Dirk Niesen, Nils Schallner

**Affiliations:** 1Department of Neurology and Neuroscience, Medical Center—University of Freiburg, 79106 Freiburg, Germany; wolf-dirk.niesen@uniklinik-freiburg.de; 2Faculty of Medicine, University of Freiburg, 79106 Freiburg, Germany; sandra.kaiser@uniklinik-freiburg.de (S.K.); matti.steimer@gmx.de (M.S.); lisa.selzner@web.de (L.S.); niels.foit@uniklinik-freiburg.de (N.A.F.); nils.schallner@uniklinik-freiburg.de (N.S.); 3Department of Anesthesiology and Critical Care, Medical Center—University of Freiburg, 79106 Freiburg, Germany; 4Department of Neurosurgery, Medical Center—University of Freiburg, 79106 Freiburg, Germany

**Keywords:** subarachnoid hemorrhage, circadian rhythm, delirium, neuro-intensive care

## Abstract

Circadian rhythm gene expression in cerebral pacemaker regions is regulated by a transcriptional-translational feedback loop across the 24-h day-night cycle. In preclinical models of subarachnoid hemorrhage (SAH), cyclic gene expression is disrupted. Stabilization of circadian rhythm gene expression attenuates susceptibility to ischemic damage in both neuronal and myocardial tissues. In this clinical observational study, circadian rhythm gene *Period-2* (*Per2*) mRNA expression levels were determined from blood leukocytes and cerebrospinal fluid (CSF) cells via real-time PCR on days 1, 7 and 14 after aneurysm rupture in 49 patients with spontaneous SAH. CSF *Per2* expression was markedly suppressed immediately after SAH and remained suppressed over the course of two weeks of ICU treatment. Short-term mortality as well as occurrence of delirium was associated with greater extent of *Per2* suppression on day 1 after SAH. Patients that developed delayed cerebral ischemia exhibited comparatively lower *Per2* expression levels on day 7 after SAH, while presence of vasospasm remained unaffected. However, *Per2* expression did not differ in patient groups with favourable or non-favourable functional neurological outcome (modified Rankin Scales 1–3 vs. 4–6). While our findings suggest a potential protective effect of stable circadian rhythm gene expression on the extent of ischemic damage, this effect was confined to the early disease course and was not reflected in patients’ functional neurological outcome.

## 1. Introduction

Spontaneous subarachnoid hemorrhage (SAH) is a subtype of hemorrhagic stroke with high mortality and morbidity. Global SAH incidence has declined from 10.2 (1980) to 6.1 (2010) per 100,000 person-years [1] and case-fatality rates have decreased by 17% between 1973 and 2002 [2], successes that have been attributed to improved cardiovascular risk factor management and more sophisticated treatment possibilities in neurocritical care [3]. However, long-term neurological disability after SAH remains substantial. Even if functional independence is regained (modified Rankin scale ≤ 3), cognitive deficits as well as mood and sleep disturbances often persist [4].

Pathophysiological processes that follow aneurysm rupture are diverse and markedly outlast the initial intracranial bleeding event, ranging from early brain injury mediated by increased intracranial pressure and consecutive global cerebral ischemia [5] to systemic and cerebral inflammatory cascades initiated by cerebral ischemia and toxicity of blood products in the subarachnoid space. Subarachnoid blood accumulation leads to activation of microglia and microglial expression of the enzyme hemoxygenase-1 (HO1), which mediates heme degradation. In addition to secondary neurological deterioration, the disease course in SAH is often aggravated by disturbances in sympathetic nerve activity mediated by the hypothalamus, causing systemic complications such as cardiomyopathy and pulmonary edema [6,7,8,9], their severity depending on the extent of neuronal injury [10]. Given the involvement of hypothalamic pathways in SAH-associated systemic pathologies, the question arises whether SAH impacts other hypothalamic functions, namely circadian rhythm control. The suprachiasmatic nucleus, located in the hypothalamus in close relation to the third ventricle and the subarachnoid space, acts as principal circadian pacemaker in mammals [11], coordinating circadian rhythm across the body and its various peripheral pacemakers according to external stimuli in a semi-autonomous way [12,13]. It exhibits oscillatory activity and gene expression across the 24 h day-night cycle that is characterized by a negative transcriptional-translational feedback loop [14]. One component of this feedback loop is the regulatory protein Period (PER), which is predominantly expressed during daytime and degraded during circadian night, generating a circadian oscillation in its own transcription by inhibiting activity of CLOCK and BMAL1 transcription factors that drive *Per* expression [15]. In communication with the SCN, other brain regions like the hippocampus as well as circumventricular organs like the choroid plexus exhibit circadian pacemaker properties [16,17]. Circadian rhythmicity has been shown to be altered in the hippocampus, as well as other brain and peripheral tissues, in response to ischemia [18]. While ischemia itself impacts circadian rhythmicity, quantitative expression of circadian genes like *Per*, *CLOCK* and *BMAL1* at the time of injury is in turn relevant for the extent of ischemic damage that is sustained by the affected tissue [19,20]. SAH inflicts not only hemorrhagic cerebral damage, especially in close proximity to CNS areas with circadian pacemaker function, but also ischemic damage both at disease onset via increased intracranial pressure, and during the disease course in the form of micro-thrombosis and delayed cerebral ischemia. In response to ischemia, *Per1*-deficient mice sustained greater neuronal damage [21], while ischemia afflicted during the peak of *Per* expression within the circadian cycle elicited less neuronal damage compared to ischemia afflicted at other times in the cycle [18,22]. Reduced expression of *Per2* heightened susceptibility to ischemic damage in the CNS after SAH and in the myocardium [23,24].

After aneurysm rupture, free heme in the subarachnoid space is degraded into biliverdin, iron and carbon monoxide (CO) by the inducible enzyme hemoxygenase-1 (HO1). Both heme and CO are critical players in the regulation of circadian gene transcriptional activity. Expression of *Per* is controlled by the transcription factors *NPAS-2* and *CLOCK*, which contain heme [25]. Activity of both transcription factors is in turn dependent on CO by means of a heme-based gas sensor mechanism [25,26]. We tested the hypothesis that *Per2* mRNA expression levels in SAH patients correlates with *HO1* expression.

We hypothesize that SAH leads to disruption of circadian rhythm gene expression as has been shown in a small patient population as well as in an animal model of SAH [22]. This study aims to further characterize alteration and temporal dynamics of *Per2* expression in response to SAH. It also aims to identify how circadian gene expression in response to SAH may impact the disease course by examining clinical parameters, namely occurrence of delirium as marker of circadian rhythm disturbance, as well as early mortality and neurological outcome after SAH. Based on preclinical data, we hypothesize that stable and elevated *Per2* expression in response to SAH exerts a neuroprotective effect which may be reflected in clinical outcome parameters.

## 2. Materials and Methods

### 2.1. Study Design

A total of 49 patients with spontaneous SAH (13 male, 36 female, age 57.6 ± 13.7 years, range 27–82 years, Table 1) were included in this study, all of whom were treated at the Intensive Care Units of the Department of Neurology and Neurophysiology and the Department of Neurosurgery at the University of Freiburg (Germany) Medical Center. The study protocol was approved by the Institutional Ethics Review Board of the University of Freiburg (Protocol No. 293/15). Informed consent from the patient, legal guardian or by proxy was provided. The trial was registered with the German Clinical Trials Register (Trial-ID DRKS00008981; Universal Trial Number U1111-1172-6077). This patient collective represents a subpopulation analysis of a concurrent study.

Inclusion criteria were: age > 18 years, spontaneous SAH confirmed either on CT scan or via lumbar puncture/presence of cerebrospinal fluid (CSF) xanthochromia, admission to the ICU, therapeutic placement of an external ventricular drain (EVD) and first CSF and blood sample collection within 24 h of ictus as well as provision of informed consent from the patient, legal guardian or by proxy.

Patients were excluded from the study if: <18 years of age, currently pregnant, admission occurred later than 24 h after ictus, there was evidence of septic aneurysm origin or evidence of ventriculitis/meningitis during the time period of sample collection or evidence of subdural/epidural hematoma on initial imaging. Patients were also excluded from the study if death occurred within 24 h of admission.

### 2.2. Sample Collection and Analysis

CSF and blood sample collections were performed during daytime (between 8 am and 5 pm) on days 1, 7 and 14 after SAH symptom onset. Blood samples were obtained from either arterial or central venous catheters, while CSF samples were acquired under sterile conditions from external ventricular drains previously placed for therapeutic purposes. Sample collection was terminated if the patients’ external ventricular drain and/or arterial/venous catheters were removed because of clinical improvement or other reasons (Figure 1).

From CSF and blood samples, *Per2* and *HO1* mRNA expression levels were determined via real-time PCR as described previously [27]:

Blood samples were stored at −80 °C in RNA stabilizing reagent tubes (Tempus Blood RNA Tube, AB#4342792). RNA was isolated from leukocytes using the correspondent spin-column RNA isolation kit (Tempus Spin RNA Isolation Kit, AB#1710145). Assessments of RNA content and purity were performed photometrically (NanoDrop 2000 Spectrophotometer, Thermo Fisher Scientific Inc., Waltham, MA, USA). RNA from CSF cells was isolated with TRIzol and concentrated by spin-column purification (RNeasy Micro Kit, Qiagen, Hilden, Germany). RNA was reversely transcribed into cDNA via reverse transcriptase PCR technique (iScript cDNA Synthesis Kit, BioRad#1708890; PeqStar 96 Universal Gradient, PeqLab). Real-time PCR (sqPCR; StepOnePlus Real Time PCR-System, A&V Applied Biosystems) with nucleic acid stain (PowerUp SYBR Green Master Mix, AB#1708020) and specific primers for *HO1*, *Per2* and *Rpl13a* (Table 2) was used for semi-quantification of cDNA. *HO1* and *Per2* constituted the target genes, while *Rpl13a* (ribosomal protein L13a) served as intra-individual reference gene. In order to calculate relative differences in mRNA expression levels, *HO1* and *Per2* expression levels from CSF and blood samples of 3 patients without intracranial hemorrhage were used as inter-individual reference population. Individual relative mRNA expression levels were calculated using the “2^−∆∆Ct^” (cycle threshold) method [28]: ∆Ct = Ct (target gene *HO1*/*Per2*) − Ct (reference gene *Rpl13a*) and ∆∆CT = ∆CT (study population) − ∆CT (reference population). 

### 2.3. Clinical Data Collection

On the basis of patients’ electronic charts and neuroimaging on admission, the following clinical parameters were recorded: demographic data (gender, age at the time of admission), subarachnoid hemorrhage severity scores (Hunt & Hess grade, modified Fisher grade, WFNS grade), Hijdra Scales (Hijdra ventricular blood score and Hijdra sum score), occurrence of delirium, vasospasm and delayed cerebral ischemia (DCI) during the time course of ICU treatment, as well as modified Rankin Scale (mRS) at admission, at discharge and one year after discharge.

Transcranial doppler sonography has been shown to have a high sensitivity (90%) to predict DCI [29]. Sonographic vasospasm was marked as present if the following criteria applied as measured by transcranial color-coded duplex sonography (TCCD): mean blood flow velocity >3 kHz (moderate)/ >4 kHz (severe) or >1 kHz increase compared to the examination on the previous day. Mean blood flow velocities were corrected for age, vertebrobasilar system, heart rate and hematocrit at the time of examination. DCI was defined as delayed onset of neurologic deterioration lasting >24 h, not explained by other causes such as electrolyte disturbance or epileptic seizures/postictal deficits or the presence of ischemic lesions on follow-up neuroimaging [30].

Functional short- and long-term outcome was assessed using the modified Rankin Scale (mRS) [31]. MRS data were collected at hospital admission and discharge as well as one year after the SAH event (assessment via telephone interview).

The first CT scan after admission to the primary healthcare facility was used to calculate radiographic total hematoma volume (OsiriX Lite software, Pixmeo, Bernex, Switzerland). First, cisternal hematoma volume (prepontine, interpeduncular and ambient cisterns) was measured from adjacent CT slices extending superiorly from the level of the caudal pons to the midbrain over a total vertical distance of 15 mm, taking slice thickness into consideration. In addition to cisternal hematoma volume, ventricular and parenchymal hematoma volumes were calculated and added if present as cisternal blood volume by itself often does not accurately represent the total intracranial hematoma burden. In addition to this quantitative assessment, the semi-quantitative Hijdra sum score [32] was determined for each patients’ initial CT scan.

### 2.4. Statistical Analyses

Statistical analyses were performed using GraphPad Prism software (version 8.4.3), GraphPad Software, San Diego, CA, USA, www.graphpad.com (accessed on 18 January 2021).

Based on previously published data [27], an a priori power analysis (Wilcoxon-Mann-Whitney-Test, group comparison CSF biomarker expression vs. functional outcome (delirium, mRS), effect size Cohen’s d 0.9; α = 0.05; power 80%) yielded a necessary sample size of 42 patients for biomarker discriminability.

Spearman non-parametrical correlation (rs, *p*) as well as linear regression (r^2^, *p*) was used to assess correlations between metrical data sets (total hematoma volume, *HO1*/*Per2* expression, modified Rankin Scale). The nonparametric, unpaired Mann-Whitney *t*-test was performed for between-group comparisons (mortality yes/no, delirium yes/no, mRS 0–3 vs. 4–6 compared to *Per2* expression) in metrical data. A mixed-effects analysis followed by Tukey’s multiple comparisons test was used to assess differences in repeated measures data (modified Rankin Scale, *Per2* expression).

Data are displayed as mean and standard deviation. In all statistical analyses, a *p*-value < 0.05 was considered to be significant.

## 3. Results

### 3.1. CSF and Blood Per2 mRNA Expression Levels Exhibit a Temporal Pattern Following SAH

On day 1 after SAH, *Per2* mRNA expression in cells isolated from CSF were suppressed compared to controls (Figure 2a, *p* = 0.0081). After the initial suppression in CSF *Per2* mRNA expression levels on day 1 after SAH, there was a non-significant increment/recovery during the first week after SAH. Expression levels were greatest on day 7 after SAH and then again decreased significantly until day 14 (day 1 vs. day 7: adjusted *p*-value = 0.230 *n* = 37, day 1 vs. day 14: adjusted *p*-value = 0.550 *n* = 12, day 7 vs. day 14: adjusted *p*-value = 0.0025 *n* = 12). However, analysis across all time points missed significance (*p* = 0.125, F (1.256, 55.88) = 2.34, Figure 2b).

At the same time, *Per2* expression in blood leukocytes showed a non-significant increase compared to healthy controls (Figure 2c, *p* = 0.059). Blood leukocyte *Per2* mRNA expression levels remained unchanged over the course of the first two weeks of ICU treatment (Figure 2d, *p* = 0.151, F (1.39, 35.59) = 2.09).

*Per2* expression levels in CSF cells were independent of gender (day 1 after SAH: *p* = 0.759, day 7 after SAH: *p* = 0.794, data not shown) and showed no consistent correlation with age (Table 3).

### 3.2. Per2 Expression Levels Are Independent of HO1 Expression

No consistent association between expression levels of *HO1* and *Per2* in CSF or blood leukocytes could be detected over the course of 14 days after the SAH event (Figure 3a, CSF *HO1* vs. *Per2* mRNA expression levels day 1 after SAH: *p* = 0.048, rs = 0.30, r^2^ = 0.011, F = 0.456, *n* = 43; day 7 after SAH: *p* = 0.556, rs = 0.1, *n* = 37). There was also no correlation between CSF *Per2* mRNA expression levels and intracranial hematoma volume as an indicator of subarachnoid heme burden (Figure 3b, day 1: *p* = 0.50, rs = 0.12, *n* = 33, day 7: *p* = 0.90, rs = 0.007, *n* = 27).

### 3.3. CSF and Blood Leukocyte Per2 Expression Levels Do Not Correlate with Clinical Outcome as Measured by Modified Rankin Scale

The patient population observed in this study showed a significant functional improvement both during the course of ICU treatment and following discharge from hospital as measured by the mRS (Figure 4a. Repeated measures mixed-effects model: *p* < 0.0001, F (1.221, 51.28) = 29.19, *n* = 37; mRS admission vs. discharge: *p* < 0.0001, *n* = 49, mRS admission vs. 1 year after discharge: *p* < 0.0001, *n* = 37, mRS discharge vs. 1 year: *p* < 0.0001, *n* = 37).

Following the hypothesis that *Per2* induction after hemorrhagic injury could exert a clinically apparent neuroprotective effect, we tested whether CSF and blood leukocyte *Per2* mRNA expression levels corresponded with clinical outcome. However, CSF *Per2* expression levels on days 1 and 7 after SAH did not differ significantly between patient groups with favorable short-term outcome (mRS at discharge 1–3: *n* = 23) vs. non-favorable short-term outcome (mRS at discharge 4–6: *n* = 20) (day 1 post SAH: *p* = 0.704, *n* = 43, day 7 post SAH: *p* = 0.557, *n* = 37, Figure 4b).

There was also no correlation between mRS 1 year after the SAH event or mRS improvement after discharge and CSF or blood leukocyte *Per2* mRNA expression levels at any of the examined timepoints (days 1,7,14 after SAH, data not shown).

### 3.4. Extent of Early Per2 Suppression May Be Associated with Higher Risk of Delirium, Delayed Cerebral Ischemia and Mortality

Patients that developed hyperactive delirium during the first week of ICU treatment (*n* = 20) showed significantly lower CSF *Per2* expression levels on day 1 compared to patients that did not develop early delirium (*n* = 14, CSF *Per2* expression (fold induction vs. control) 0.17 ± 0.08 vs. 0.12 ± 0.12, *p* = 0.008, Figure 4c). Interestingly, delirium that occurred later during ICU treatment showed no association with lower CSF *Per2* expression (*Per2* expression day 7 vs. delirium week 2: *n* = 26, *p* = 0.634). Blood leukocyte *Per2* expression levels were largely equal in patient subgroups with or without delirium (day 1 after SAH vs. delirium week 1: *p* = 0.967, *n* = 40, day 7 after SAH vs. delirium week 2: *p* = 0.274, *n* = 29).

Further analyzing different patient subgroups, patients that died during hospital treatment showed significantly lower CSF *Per2* mRNA expression levels on day 1 after SAH than those that survived the first weeks after the SAH event (*p* = 0.017, *n* = 43, Figure 4d). In contrast, the following *Per2* expression levels displayed no more association with mortality rates (CSF *Per2* expression day 7 vs. mortality yes/no: *p* = 0.676, *n* = 37). Blood leukocyte *Per2* expression on days 1 and 7 after SAH also did not show differences regarding early mortality (day 1: *p* = 0.194, *n* = 48, day 7: *p* > 0.99, *n* = 38, data not shown).

There was no association between occurrence of sonographic vasospasm over the course of ICU treatment and CSF *Per2* expression levels (Figure 5a). However, patients that developed delayed cerebral ischemia (DCI) at any point in time exhibited lower CSF *Per2* expression levels on day 7, but not on days 1 and 14 after SAH (Figure 5b, day 7: *p* = 0.012, *n* = 35, day 1: *p* = 0.791, *n* = 39, day 14: *p* = 0.92, *n* = 11).

## 4. Discussion

Our data showed a disturbance of circadian gene expression mainly in CSF cells and to a lesser extent in blood leukocytes, which was reproducible across different time-points over two weeks after the SAH event. However, while the disturbance compared to controls was apparent very early (day 1) after the hemorrhagic event, *Per2* expression levels in CSF and blood did not change significantly over the course of the following two weeks. *HO1* expression, an indicator of subarachnoid blood burden and disease severity, showed no correlation with *Per2* expression levels. CSF and blood leukocyte *Per2* expression levels were also unable to predict functional short- and long-term outcome. However, there was an association between the extent of *Per2* expression on day 1 after SAH and the incidence of early hyperactive delirium and mortality, indicating possible detrimental effects of early circadian rhythm disruption after SAH. Furthermore, sustained suppression of CSF *Per2* expression one week after SAH was associated with higher incidence of delayed cerebral ischemia (summarized in Table 4).

In this study, SAH patients showed a marked suppression of *Per2* expression in the CSF and non-significant upregulation of *Per2* expression in blood leukocytes as early as day 1 after the bleeding event. These findings are in contrast to previously published work that described an upregulation of *Per2* expression in CSF cells after SAH, while *Per2* expression in blood leukocytes remained unchanged compared to controls [22]. Nonetheless, both suppression of *Per2* expression in the CSF and upregulation in blood leukocytes indicate a disruption of circadian rhythm gene expression after SAH compared to control individuals without intracranial hemorrhage. Our study featured a larger sample size (49 vs. 13 patients included), which could account at least in part for the observed differences in expression levels. It should also be taken into account that while CSF and blood samples were both taken at the same time for one patient on the specified days, the time of day at which samples were collected was not standardized across the daytime (8 a.m. to 5 p.m.) and varied between patients. As *Per2* expression in the SCN is upregulated during the circadian day, the time of day at which samples were collected could influence expression levels and thus presents a relevant limitation of this study. While sample collection was performed on days 1, 7 and 14 after SAH in each patient, it was not possible to control for the exact daytime and the number of hours between the SAH event and sample collection. Future studies should take this into account and examine intra-individual expression levels across the day- and night-time and in relation to the time of day at which the hemorrhage occurred.

The observed suppression of *Per2* expression in the CSF could be an indicator of desynchronization of different circadian oscillators within the CNS. Besides the SCN there are other independent brain regions that exhibit circadian rhythmicity [16], and desynchronization of these oscillators perpetuated by the SAH event could mimic *Per2* suppression in the CSF. Recently, the choroid plexus has been identified as one of the peripheral circadian clocks in rat models, exhibiting robust oscillatory expression of *Per2* as well as signaling to the SCN [17,33]. The choroid plexus lines the ventricle walls and produces cerebrospinal fluid by filtration of blood plasma. It should be considered that *Per2* expression in the CSF may reflect circadian clock (dys) function of the choroid plexus rather than the SCN, the main circadian coordinator.

Transcriptional activity of NPAS-2 and CLOCK, transcription factors that control expression of *Per*, has been shown to be both heme- and CO-dependent [25,26,34]. Both substances are in turn released by heme degradation that is mediated by the enzyme hemoxygenase-1 (HO1). HO1 enzyme activity in turn is a function of intracranial hematoma burden and thus SAH severity [27]. We questioned whether the extent of hemorrhagic brain injury, i.e., heme burden, could influence *Per2* expression and could thus determine the extent of circadian gene disruption. There was, however, no consistent association between blood or CSF *HO1* expression levels or subarachnoid hematoma volume and *Per2* expression. These results indicate that heme-dependent transcriptional regulation of circadian genes may not be determined by the quantitative heme burden or HO1 enzyme activity. The patient population observed in this study consisted of mostly poor-grade SAH patients (Hunt & Hess grade 3.22 ± 1.18) with significant subarachnoid hematoma burden (24.65 ± 24.99 cm^3^), so that possible ceiling effects of transcriptional regulation mediated by heme and CO availability should be taken into consideration. Regulation and expression of circadian genes is also customarily assessed in neuronal tissues such as the SCN and hippocampus in animal models [16,35]. In the clinical setting however, CNS circadian gene expression can only be assessed from CSF, although preclinical or comparable data is lacking with regard to circadian gene expression quantification. While it has been asserted that microglia upregulate *HO1* expression in response to subarachnoid hemorrhage [36] and this upregulation can be observed in neuronal tissue as well as human CSF [37], there have been no studies yet that correlate neuronal tissue *Per2* expression with availability in CSF. Due to limited amounts of biological material, it was not possible to identify the cell types responsible for circadian rhythm gene expression in human CSF, which constitutes a limitation when interpreting the results of this study. As mentioned earlier, it is possible that CSF *Per2* expression is a reflection of choroid plexus *Per2* expression and disruption of its role as circadian pacemaker. The majority of patients in this study had significant ventricular hematoma volumes (4.92 ± 14.75 cm^3^) as well as acute hydrocephalus on admission (*n* = 37 vs. *n* = 6 without acute hydrocephalus) which implied disturbance of CSF circulation and necessitated the placement of external ventricular drains. Suppression of CSF *Per2* expression could present an interesting downstream complication of both direct blood component toxicity and elevated intraventricular pressure on the choroid plexus/periventricular tissue.

In models of SAH as well as traumatic brain injury, circadian clock gene expression and locomotor activity was altered in response to injury [22,35]. In this study, the patient group that showed clinically manifest delirium over the first week of ICU treatment had comparatively lower *Per2* expression levels in the CSF on day 1 after SAH. As *Per2* expression in the CSF overall was suppressed in comparison to controls, this could indicate that higher CSF *Per2* expression results in lesser circadian rhythm disruption and thus lower incidence of delirium. As observed differences between the patient groups were small and could only be observed during week 1 after SAH, these results need to be interpreted with caution and require further validation.

There was no association between CSF or blood *Per2* expression levels and functional short- or long-term outcome. As determination of mRNA expression from the CSF may not reflect neuronal tissue *Per2* expression or may be an indicator of choroid plexus circadian gene expression as discussed above, it may not be possible to directly corroborate preclinical findings of *Per2*-mediated neuroprotection. It should also be considered that the patient collective was clinically homogenous in the sense that all patients were severely affected (mRS on admission 4.3 ± 0.96). Disability resulting from immediate early brain injury and consecutive neuroinflammatory processes leading to secondary neuronal injury after SAH were likely too severe for interindividual differences in CSF *Per2* expression to show an association with functional outcome.

Animal models have demonstrated increased susceptibility to ischemic myocardial injury with disturbed glycogen metabolism and increased inflammation in *Per2* knockout animals [23,38]. On the other hand, increasing circadian *Per2* amplitude via intense light exposure facilitated cellular adaptation to myocardial ischemia and regulated endothelial barrier function during hypoxia [39]. The observation that delayed cerebral ischemia as well as early mortality after SAH were associated with relatively lower CSF *Per2* expression levels on day 7/day 1 respectively could support the existing preclinical data demonstrating *Per2*-dependent cardio- and neuroprotection from ischemia [22,23]. The observation that the occurrence of DCI was associated with lower *Per2* expression on day 7 is especially interesting considering that *Per2* expression was equal in patient subgroups with or without vasospasm. This may indicate an independent effect of *Per2* expression on ischemia tolerance. Despite the association of early lower *Per2* expression with mortality and DCI, no effect was observed regarding functional neurological outcome. It is likely that secondary, especially neuroinflammatory complications after SAH are too severe to allow for the observed small differences in ischemic complications to become evident in outcome scores. With regards to the time course of *Per2* expression, it is possible that early *Per2* induction during the first week after aneurysm rupture is responsible for protection against ischemia and early brain injury in SAH, while dynamics in *Per2* expression that occur later during the course of the disease may not be as relevant—however, the data as of yet do not allow this conclusion.

While our findings suggest a potential protective effect of stable circadian rhythm gene expression on the extent of ischemic damage and are thus in line with preclinical data, this effect was confined to the early disease course. Considering overall mortality and functional neurological outcome, *Per2* expression levels did not differ between patient subgroups.

## 5. Conclusions

This study showed disruption of circadian rhythm gene *Per2* expression in CSF and blood leukocytes in patients with spontaneous SAH. As there are multiple circadian pacemakers in the CNS besides the main coordinator, the SCN, at this stage it cannot be distinguished which CNS regions may be reflected in CSF *Per2* expression. While preclinical data has shown circadian gene expression to be both heme- and CO-dependent, in this clinical study CSF *Per2* expression exhibited no correlation with subarachnoid heme burden or expression of the heme-catalyzing enzyme HO1. This study found a possible association between the extent of early *Per2* suppression after SAH and early delirium incidence, delayed cerebral ischemia and mortality, but not with functional neurological outcome.

## Figures and Tables

**Figure 1 life-11-00124-f001:**
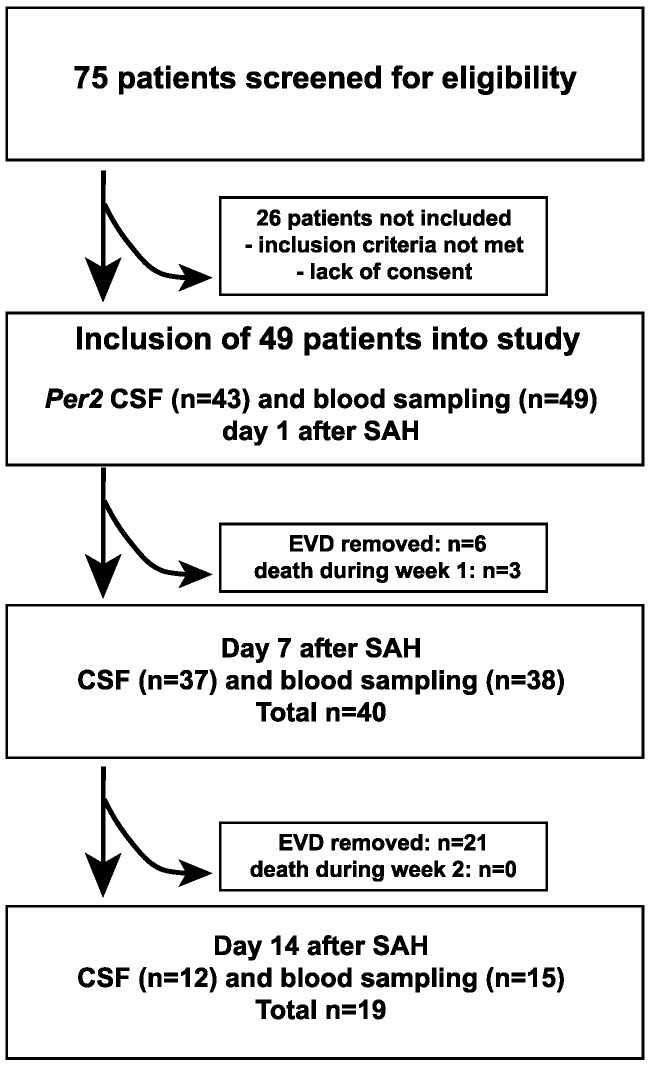
Schematic depiction of the study protocol.

**Figure 2 life-11-00124-f002:**
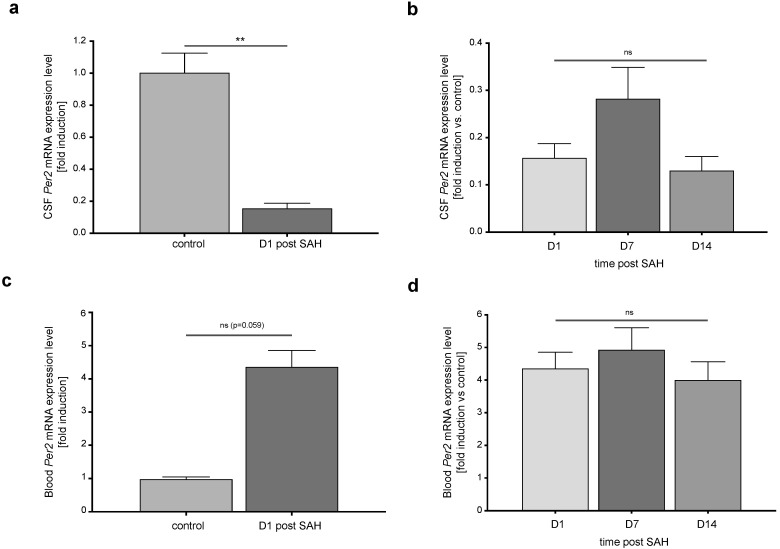
*Per2* mRNA expression levels in CSF cells and blood leukocytes are altered after SAH. (**a**) CSF *Per2* mRNA expression levels were suppressed as early as day 1 after SAH compared to controls (*n* (controls) = 2, *n* = (SAH patients D1 post SAH) = 43, *p* = 0.008). (**b**) Over the course of ICU treatment, CSF *Per2* expression levels remained unchanged in SAH patients (*p* = 0.125, F (1.256, 55.88) = 2.34). (**c**) Blood leukocyte *Per2* mRNA expression in SAH patients were non-significantly increased compared to controls (*n* (controls) = 2, *n* (SAH patients) = 48, *p* = 0.059). (**d**) Blood leukocyte *Per2* mRNA expression remained stable over the course of the following weeks (*p* = 0.151, F (1.39, 35.59) = 2.09). CSF: cerebrospinal fluid, *Per2*: Period-2, SAH: subarachnoid hemorrhage. Data presented as mean + SEM, ** *p* < 0.01.

**Figure 3 life-11-00124-f003:**
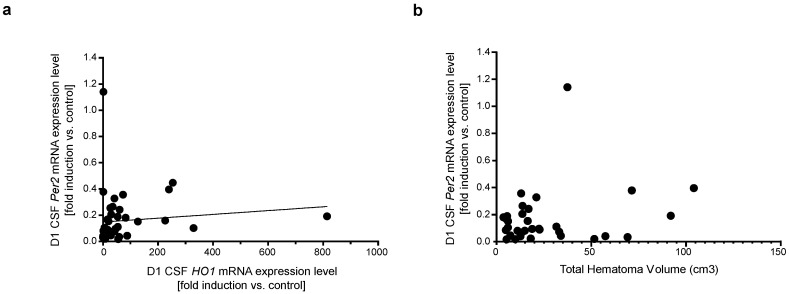
CSF *Per2* expression levels are largely independent of *HO1* expression and subarachnoid hematoma burden. (**a**) CSF *HO1* vs. *Per2* expression levels on day 1 after SAH (*p* = 0.048, rs = 0.30, r^2^ = 0.011, F = 0.456, *n* = 43). (**b**) *Per2* mRNA expression showed no correlation with intracranial hematoma volume (*p* = 0.50, rs = 0.12, *n* = 33). CSF: cerebrospinal fluid, *HO1*: hemoxygenase-1, *Per2*: Period-2, SAH: subarachnoid hemorrhage.

**Figure 4 life-11-00124-f004:**
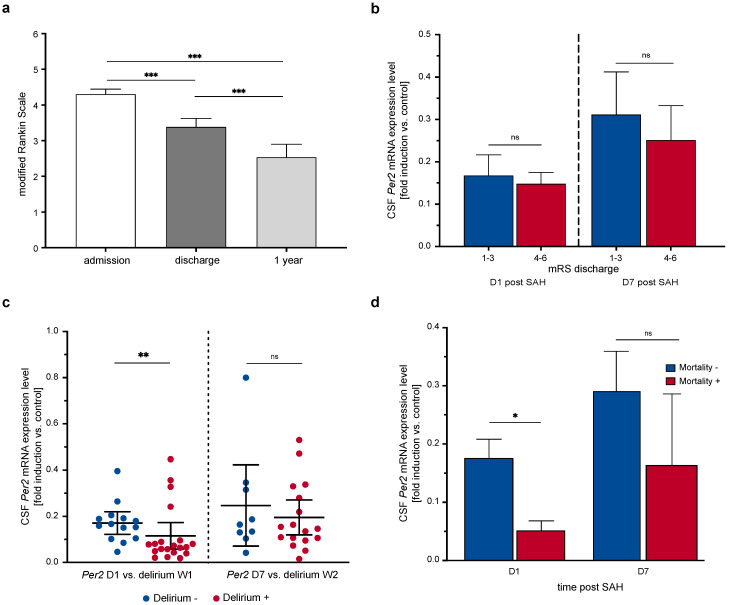
Extent of early CSF *Per2* suppression is associated with delirium and early mortality, but not functional outcome. (**a**) Functional short- and long-term improvement after SAH reflected in mRS scores at admission, discharge from hospital and 1 year after the SAH event (*p* < 0.0001, F (1.221, 51.28) = 29.19, *n* = 37). (**b**) CSF *Per2* expression levels did not differ between patient groups with favorable (mRS 1–3, *n* = 23)) and non-favorable (mRS 4–6, *n* = 20) clinical outcome at discharge from hospital (day 1: *p* = 0.704, day 7: *p* = 0.557). (**c**) Early, but not late delirium was associated with higher CSF *Per2* expression (*p* = 0.008, *n* = 34). (**d**) Short-term mortality was associated with lower CSF *Per2* expression levels on day 1, but not day 7 after SAH (day 1: *p* = 0.017, *n* = 43 (*n* = 37 survivors), day 7: *p* = 0.676, *n* = 37 (*n* = 35 survivors). CSF: cerebrospinal fluid, mRS: modified Rankin Scale, *Per2*: Period-2, SAH: subarachnoid hemorrhage, W: week. Data presented as mean + SEM/mean ± 95% confidence interval (**c**), * *p* < 0.05, ** *p* < 0.01, *** *p* < 0.001.

**Figure 5 life-11-00124-f005:**
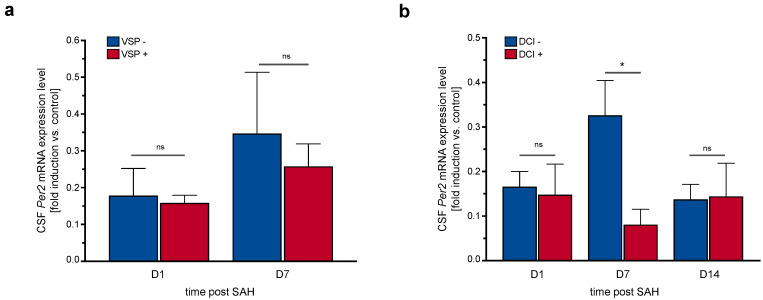
CSF *Per2* expression levels cannot predict occurrence of vasospasm but may be associated with incidence of DCI. (**a**) CSF *Per2* expression levels on days 1 and 7 after SAH did not differ between patient subgroups with or without sonographic vasospasm (day 1: *p* = 0.119, *n* = 40, day 7: *p* = 0.572, *n* = 35). (**b**) Occurrence of delayed cerebral ischemia was associated with lower CSF *Per2* mRNA expression levels on day 7, but not days 1 and 14 after SAH (day 7: *p* = 0.012, *n* = 35, day 1: *p* = 0.791, *n* = 39, day 14: *p* = 0.92, *n* = 11). CSF: cerebrospinal fluid, DCI: delayed cerebral ischemia, *Per2*: Period-2, SAH: subarachnoid hemorrhage, VSP: vasospasm. Data presented as mean + SEM, * *p* < 0.05.

**Table 1 life-11-00124-t001:** Clinical characteristics of the study population.

	Mean ± SD	*n*
Age	57.63 ± 13.71	49
Modified Rankin Scale		
on admission	4.30 ± 0.96	49
at discharge	3.39 ± 1.64	49
1 year after SAH	2.54 ± 2.17	37
SAH severity scales		
Hunt & Hess grade	3.22 ± 1.18	49
WFNS grade	3.32 ± 1.43	47
Modified Fisher grade	3.65 ± 0.67	46
Total hematoma volume (cm^3^)	24.65 ± 24.99	38
Hijdra Sum Score	23.67 ± 9.52	45

SAH: subarachnoid hemorrhage, SD: standard deviation, WFNS: World Federation of Neurosurgical Societies.

**Table 2 life-11-00124-t002:** Primer sequences used in cerebrospinal fluid (CSF) and blood real-time PCR analyses.

	Primer Sequence
*HO1* forward	GTGATAGAAGAGGCCAAGACTG
*HO1* reverse	GAATCTTGCACTTTGTTGCTGG
*Per2* forward	TCCTCGGCTTGAAACGGC
*Per2* reverse	GAACGAAGCTTTCGGACCTCA
*Rpl13a* forward	CGGACCGTGCGAGGTAT
*Rpl13a* reverse	CACCATCCGCTTTTTCTTGTC

*HO1*: hemoxygenase-1, *Per2*: Period-2, *Rpl13a*: ribosomal protein L13a.

**Table 3 life-11-00124-t003:** Correlation of *Per2* mRNA expression levels and patients’ age.

*Per2* mRNA Expression Levels [Fold Induction vs. Control]	Spearman r	*p*-Value	*n*
CSF			
Day 1	−0.254	0.100	43
Day 7	0.014	0.932	37
Day 14	−0.081	0.804	12
Blood			
Day 1	0.290	0.046	48
Day 7	0.162	0.330	38
Day 14	0.588	0.018	16

CSF: cerebrospinal fluid; *Per2*: Period-2.

**Table 4 life-11-00124-t004:** Summary.

CSF *Per2* mRNA Expression Level [Fold Induction vs. Control]	Day 1			Day 7		
	Mean ± SD	*n*	*p*-Value	Mean ± SD	*n*	*p*-Value
Modified Rankin Scale discharge						
1–3	0.167 ± 0.236	23	0.763	0.311 ± 0.452	20	0.557
4–6	0.148 ± 0.122	20		0.250 ± 0.337	17	
Vasospasm						
negative	0.177 ± 0.290	15	0.119	0.346 ± 0.580	12	0.572
positive	0.157 ± 0.112	25		0.257 ± 0.295	23	
Delayed cerebral ischemia						
negative	0.165 ± 0.204	34	0.791	0.325 ± 0.431	30	0.012
positive	0.148 ± 0.154	5		0.079 ± 0.079	5	
Delirium						
negative	0.171 ± 0.085	14	0.008	0.247 ± 0.229	9	0.634
positive	0.115 ± 0.123	20		0.195 ± 0.147	17	
Mortality						
negative	0.175 ± 0.199	37	0.017	0.290 ± 0.408	35	0.675
positive	0.051 ± 0.041	6		0.163 ± 0.173	2	

CSF: cerebrospinal fluid; *Per2*: Period-2.

## Data Availability

The data presented in this study are available on request from the corresponding author.

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
