# Peer review of "Patients with Subarachnoid Hemorrhage Exhibit Disturbed Expression Patterns of the Circadian Rhythm Gene Period-2"

_life, 2021, doi:10.3390/life11020124_

Round 1
Reviewer 1 Report
The manuscript is very interesting and well written/argued. Only one observation - the bibliography could be improved in terms of adding citations from the last 2-3 years.
Author Response
We express our appreciation to the Reviewer for the time and effort spent in reviewing our work. We very much hope that the following replies and revisions of the manuscript have appropriately addressed the Reviewers’ concerns.
Reviewer’s comments:
1.) Reviewer: The manuscript is very interesting and well written/argued. Only one observation - the bibliography could be improved in terms of adding citations from the last 2-3 years.
Reply: We updated the bibliography with the following more recent publications:
Kagiyama, N.; Sugahara, M.; Crago, E.A.; Qi, Z.; Lagattuta, T.F.; Yousef, K.M.; Friedlander, R.M.; Hravnak, M.T.; Gorcsan, J. Neurocardiac Injury Assessed by Strain Imaging Is Associated With In-Hospital Mortality in Patients With Subarachnoid Hemorrhage. JACC: Cardiovascular Imaging 2020, 13, 535–546, doi:10.1016/j.jcmg.2019.02.023.
Gottlieb, E.; Landau, E.; Baxter, H.; Werden, E.; Howard, M.E.; Brodtmann, A. The Bidirectional Impact of Sleep and Circadian Rhythm Dysfunction in Human Ischaemic Stroke: A Systematic Review. Sleep Medicine Reviews 2019, 45, 54–69, doi:10.1016/j.smrv.2019.03.003.
Qiu, Z.; Ming, H.; Lei, S.; Zhou, B.; Zhao, B.; Yu, Y.; Xue, R.; Xia, Z. Roles of HDAC3-Orchestrated Circadian Clock Gene Oscillations in Diabetic Rats Following Myocardial Ischaemia/Reperfusion Injury. Cell Death Dis2021, 12, 43, doi:10.1038/s41419-020-03295-y.
Reviewer 2 Report
The authors tested the expression of the circadian rhythm gene Period-2 in SAH patients, and found that Period-2 mRNA expression is significantly suppressed after SAH. However, Period-2 expression is not associated with the functional neurological outcome. Some specific concerns need to be addressed.
Specific concerns:
1. Period-2 is a circadian rhythm gene. It would be very important to measure the Period-2 expression at the samples collected at the same time point after SAH, although it is impossible to do so. This issue should be mentioned or discussed.
2. Both male and female patients with the age range of 27-82 years were included in this study. Are there significant differences in the expression of Period-2 in patients of different ages and genders? It would be helpful to check the impact of sex and age on Period-2 expression after SAH.
3. What are the cells in CSF and Blood that mainly express Period-2 after SAH?
Author Response
We express our appreciation to the Reviewer for the time and effort spent in reviewing our work. We very much hope that the following replies and revisions of the manuscript have appropriately addressed the Reviewers’ concerns.
Reviewer’s comments:
1.) Reviewer: Period-2 is a circadian rhythm gene. It would be very important to measure the Period-2 expression at the samples collected at the same time point after SAH, although it is impossible to do so. This issue should be mentioned or discussed.
Reply: We agree with the reviewer that expression of circadian genes is time sensitive and the timing of sample collection might impact the results presented in our manuscript. In a clinical setting, timing all sample collections at the exact same time point is impossible to accomplish. However, we collected all samples at daytime, an information that was added to the methods section. The section now reads:
CSF and blood sample collections were performed during daytime (between 8 am and 5 pm) on days 1, 7 and 14 after SAH symptom onset. (page 3, lines 37-38)
We also added a section to the discussion to properly acknowledge this limitation of our study. The section now reads:
As Per2 expression in the SCN is upregulated during the circadian day, the time of day at which samples were collected could influence expression levels and thus presents a relevant limitation of this study. While sample collection was performed on days 1, 7 and 14 after SAH in each patient, it was not possible to control for the exact daytime and the number of hours between the SAH event and sample collection. Future studies should take this into account and examine intra-individual expression levels across the day- and night-time and in relation to the time of day at which the hemorrhage occurred. (page 10 line 29 – page 11 line 3)
However, we know from pre-clinical murine SAH models that the naturally occurring oscillating expression of circadian genes is abolished following SAH (Schallner et al. Stroke 2017; 48:2565–2573). It is tempting to speculate that this will also occur in human SAH and that the time of day after SAH will not influence the extent of circadian gene expression in the same way as in humans without cerebral hemorrhage.
2.) Reviewer: Both male and female patients with the age range of 27-82 years were included in this study. Are there significant differences in the expression of Period-2 in patients of different ages and genders? It would be helpful to check the impact of sex and age on Period-2 expression after SAH.
Reply: We agree that age and sex are two important co-variables that might influence the expression of Per-2 at baseline and after injury. We have previously analyzed, whether Per-2 expression correlates with sex and age and since it did not show any significant interdependence, we didn’t report these results. The correlation analysis for Per2/age is now shown as Table 3 on page 7. Per2 expression levels did not differ significantly between male and female patient groups (page 6 lines 21-23).
3.) Reviewer: What are the cells in CSF and Blood that mainly express Period-2 after SAH?
Reply: The reviewer raises a very important question that we weren’t able to solve during several years of research related to this matter. From previous studies (J Clin Invest. 2015; 125:2609–2625; Antioxidants 2019; 8, 496) we know that the cells mainly contained in the CSF (and therefore analyzed) are of myeloid origin. However, the exact differentiation between potentially infiltrating bone-marrow derived macrophages and resident microglia remains challenging due to the lack of specific makers and due to the scarcity of biological material obtained during our studies. The question is important since it remains an ongoing scientific controversy whether resident microglia re-generate and multiply from specific progenitors within in the brain or whether the population is fed from peripheral bone-marrow derived myeloid cells and what cell type mainly contributes to neuroprotection following hemorrhage. Some evidence from other groups and our own work (J Clin Invest. 2015; 125:2609–2625) suggest that it is mainly the microglia population responsible and that infiltrating leucocytes are not of the same relevance. However, it remains to be proven that this is also true in human SAH. In the future we will aim to approach this question with new techniques that will allow us characterize the cells contained in the CSF with gene sequencing, for example.
We also added a section to the discussion to properly acknowledge this limitation of our study. The section now reads:
Due to limited amounts of biological material, it was not possible to identify the cell types responsible for circadian rhythm gene expression in human CSF, which constitutes a limitation when interpreting the results of this study (page 11, lines 35-37).
Reviewer 3 Report
first of all thank the revision of the text.
The article presented is of interest despite the fact that the sample is not very large.
In my opinion, the article should follow the template of the magazine and the methodology should be placed after the introduction.
on the other hand, the presentation of the results is not easy to read. I doubt if the graphs provide insight. I recommend tables with means and standard deviation, as they make understanding easier.
The introduction is of interest and well written, I would include an article that addresses the time of care in stroke, I recommend the following article in this paragraph:
"successes that have been attributed to improved cardiovascular risk factor management and more sophisticated treatment possibilities in neurocritical care"
https://doi.org/10.3390/jcm8101712
therefore, the article is well written, and it is necessary to modify the structure, and add tables, eliminating the graphics that do not provide knowledge.
Author Response
We express our appreciation to the Reviewer for the time and effort spent in reviewing our work. We very much hope that the following replies and revisions of the manuscript have appropriately addressed the Reviewers’ concerns.
Reviewer’s comments:
1.) Reviewer: In my opinion, the article should follow the template of the magazine and the methodology should be placed after the introduction.
Reply: We corrected the sequence of paragraphs. The methods are now placed after the introduction.
2.) Reviewer: On the other hand, the presentation of the results is not easy to read. I doubt if the graphs provide insight. I recommend tables with means and standard deviation, as they make understanding easier.
Reply: We believe that the graphic presentation of our data helps with understanding of the results. However, to meet the reviewer’s criticism, we also added a tabulated summary of our results, as can now be seen as Table 4.
3.) Reviewer: The introduction is of interest and well written, I would include an article that addresses the time of care in stroke, I recommend the following article in this paragraph: "successes that have been attributed to improved cardiovascular risk factor management and more sophisticated treatment possibilities in neurocritical care" https://doi.org/10.3390/jcm8101712.
Reply: We added the citation as recommended by the reviewer.
Round 2
Reviewer 3 Report
The authors have made the suggested changes, publication is possible.